# Pregnant women and male partner perspectives of secondary distribution of HIV self-testing kits in Uganda: A qualitative study

**Michelle A. Bulterys**[1,2]*, **Brienna Naughton**[2], **Andrew Mujugira**[2,3], **Jackson Mugisha**[3], **Agnes Nakyanzi**[3], **Faith Naddunga**[3], **Jade Boyer**[2], **Norma Ware**[4], **Connie Celum**[2], **Monisha Sharma**[1,2]

**1** Department of Epidemiology, University of Washington, Seattle, Washington, United States of America, **2** Department of Global Health, University of Washington, Seattle, Washington, United States of America, **3** Infectious Diseases Institute, Makerere University, Kampala, Uganda, **4** Department of Global Health and Social Medicine, Harvard Medical School, Boston, Massachusetts, United States of America

* mbult@uw.edu

## Abstract

### Background

HIV self-testing (HIVST) is a promising strategy to increase awareness of HIV status among sub-Saharan African (SSA) men. Understanding user perspectives on HIVST secondary distribution from pregnant women attending antenatal care (ANC) to their male partners is crucial to optimizing delivery strategies.

### Methods

We sampled pregnant women attending ANC without their partners and purposively over-sampled pregnant women living with HIV (PWHIV) to understand their unique views. We recruited male partners after obtaining contact information from women. We conducted 14 focus group discussions and 10 in-depth interviews with men and pregnant women. We assessed acceptability of HIVST secondary distribution, barriers, facilitators, and interventions to increase HIVST uptake.

### Results

Participants felt that HIVST secondary distribution was acceptable, particularly for women in stable relationships. However, many expressed concerns about accusations of mistrust, relationship dissolution, fear of discovering serodifference, and lack of counseling associated with HIVST. PWHIV reported hesitation about secondary distribution, citing fears of unintended HIV status disclosure and abandonment resulting in financial hardship for themselves and their infant. Some participants preferred that providers contact men directly to offer HIVST kits instead of distribution *via* women. Participants reported that community sensitization, availability of phone-based counseling, male clinic staff, extended clinic hours, and financial incentives could increase men's HIVST use and linkage to care.

**Data Availability Statement:** All relevant data are within the paper and its Supporting information files.

**Funding:** This study was funded by the U.S. National Institute of Mental Health (K01MH115789), awarded to Dr. Monisha Sharma. The funders had no role in study design, data collection, analysis, writing of the report, nor the decision to submit for publication.

**Competing interests:** The authors report no conflicts of interest.

## Conclusion

Participants expressed high interest in using HIVST, but secondary distribution was not universally preferred. We identified potential strategies to increase HIVST acceptability, particularly among PWHIV and those in unstable partnerships which can inform strategies to optimize HIVST distribution.

## Introduction

Compared to women, men in sub-Saharan Africa (SSA) have suboptimal rates of HIV testing, resulting in delayed initiation of antiretroviral therapy (ART) and increased morbidity, mortality, and onward transmission [1]. In Uganda, approximately 1.4 million people were living with HIV in 2021, of whom 91% were aware of their status [2]. High ANC attendance in SSA (93% in Uganda) results in the majority of women receiving integrated HIV testing services during pregnancy [3]. However, men's HIV testing rates remain low (73%) despite efforts to improve testing uptake (e.g. invitation letters for fast-track testing at ANC) [4].

HIV self-testing (HIVST) is shown to achieve high uptake among men in SSA and could increase men's HIV testing and linkage to care [5–14]. A promising HIVST delivery strategy is secondary distribution, whereby pregnant women attending antenatal care (ANC) are given an HIVST kit to deliver to their male partners and are trained on its use and interpretation (Fig 1a) [15, 16]. Among pregnant women living with HIV (PWHIV), HIVST secondary distribution can promote couples testing and disclosure which can increase women's retention in ART and prevention of mother-to-child transmission (PMTCT) programs [17]. Furthermore, testing partners of PWHIV can be a high yield strategy to identify men with HIV [18, 19].

The World Health Organization (WHO) has recommended scale-up of secondary distribution of HIVST kits to pregnant women to give to their partners; several countries in SSA, including Uganda, have begun national rollout [20, 21]. Because HIVST is a screening test, Uganda Ministry of Health Guidelines encourage individuals who self-test positive to seek HIV clinical services for confirmatory testing and linkage to treatment depending on results [22]. Studies have shown high uptake of HIVST among men after secondary distribution. However, most studies relied on self-report from female partners and the vast majority of pregnant women were HIV-negative [6]. There are little qualitative data regarding women and men's perspectives on implementation strategies to increase women's HIVST distribution, and men's testing uptake and clinic linkage. Further, there is a lack of data assessing perspectives of PWHIV who face unique barriers including fear of HIV status disclosure. Finally, the success of HIVST distribution is contingent on linkage to care among men who self-test HIV-positive; however, data are lacking on successful interventions to increase clinic linkage. We assessed pregnant women and male partners' perspectives regarding HIVST acceptability, barriers, facilitators, and strategies to optimize HIVST secondary distribution and men's clinic linkage.

## Methods

### Study design and participant recruitment

We conducted focus group discussions (FGDs) and in-depth interviews (IDIs) between 2019–2020. Pregnant women reporting not knowing their partner's HIV status were recruited in person from two public ANC clinics in Kampala, Uganda. Additional women who were not invited to participate in FGDs and IDIs were asked to provide phone numbers for their partners. A male Ugandan qualitative researcher and qualified translator (JM) telephoned partners

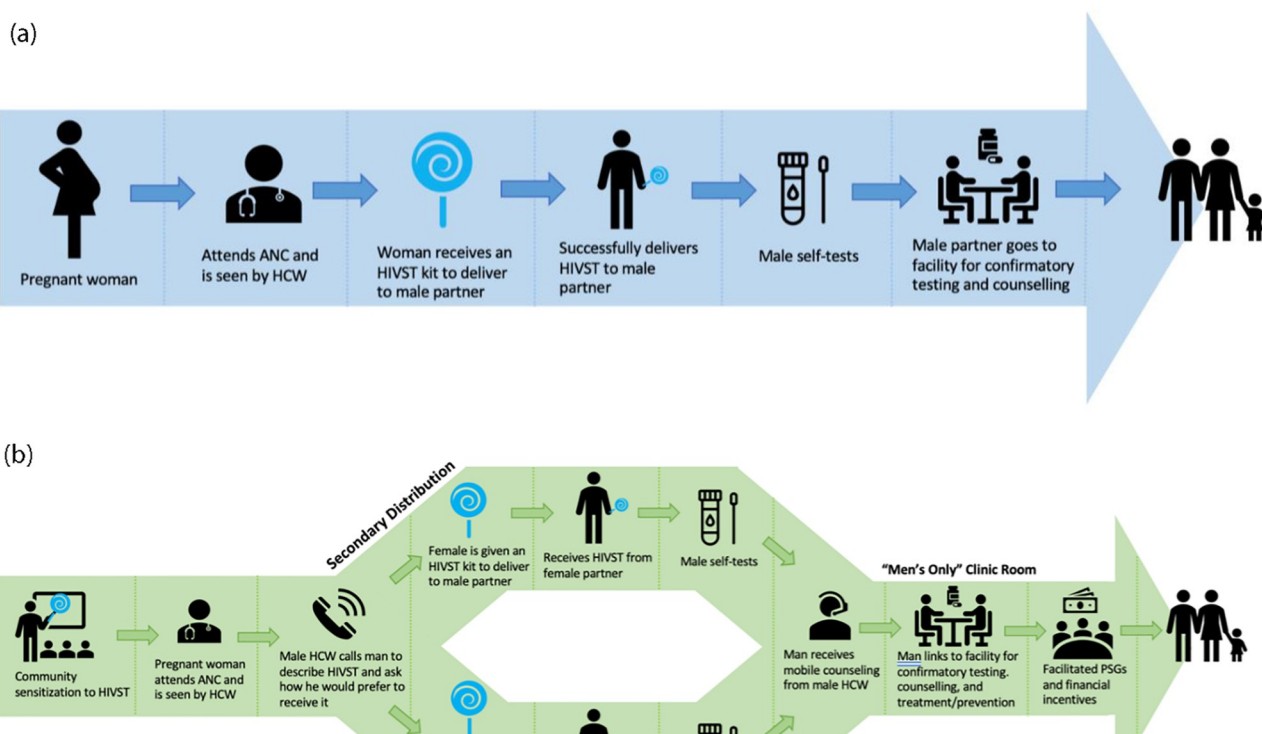

**Fig 1. a.** Standard cascade of HIVST secondary distribution. **b.** Re-envisioned cascade of HIVST distribution*. *PSG: peer support groups.

of consenting women to invite them to participate. Eligibility criteria for all participants were: ≥18 years and willing to provide informed consent. Eligibility criteria for women were: currently pregnant, in a relationship, not aware of their partner's HIV status, not attending ANC with their partner, and low risk for intimate partner violence (IPV). Healthcare workers assessed women's IPV risk following the World Health Organization's standardized screening tool for clinical diagnosis of IPV [23]. Eligibility criteria for men were: having a pregnant partner attending ANC and having a working phone for contact purposes.

FGDs had 8–12 participants per group to include multiple perspectives on barriers and facilitators to HIVST uptake and clinic linkage [11, 24]. Separate FGDs were held for men and women; among women, FGDs were separated by HIV status, aside from one. We purposively sampled three FGDs consisting of only PWHIV to provide a safe space and understand how their perspectives differed from HIV-negative women. IDIs were conducted by JM alone with men and women not participating in FGDs to further explore personal narratives regarding HIV testing, HIVST delivery, and disclosure. Since HIVST secondary distribution was not part of standard ANC care at the time of data collection, participants were asked to speak hypothetically about their willingness to use HIVST.

## Data collection and analysis

Semi-structured interview topics included HIV risk perception; gender norms; barriers and facilitators of facility HIV testing, secondary distribution of HIVST, couples testing and disclosure, and clinic linkage; and interventions to increase HIVST uptake and clinic linkage [24,

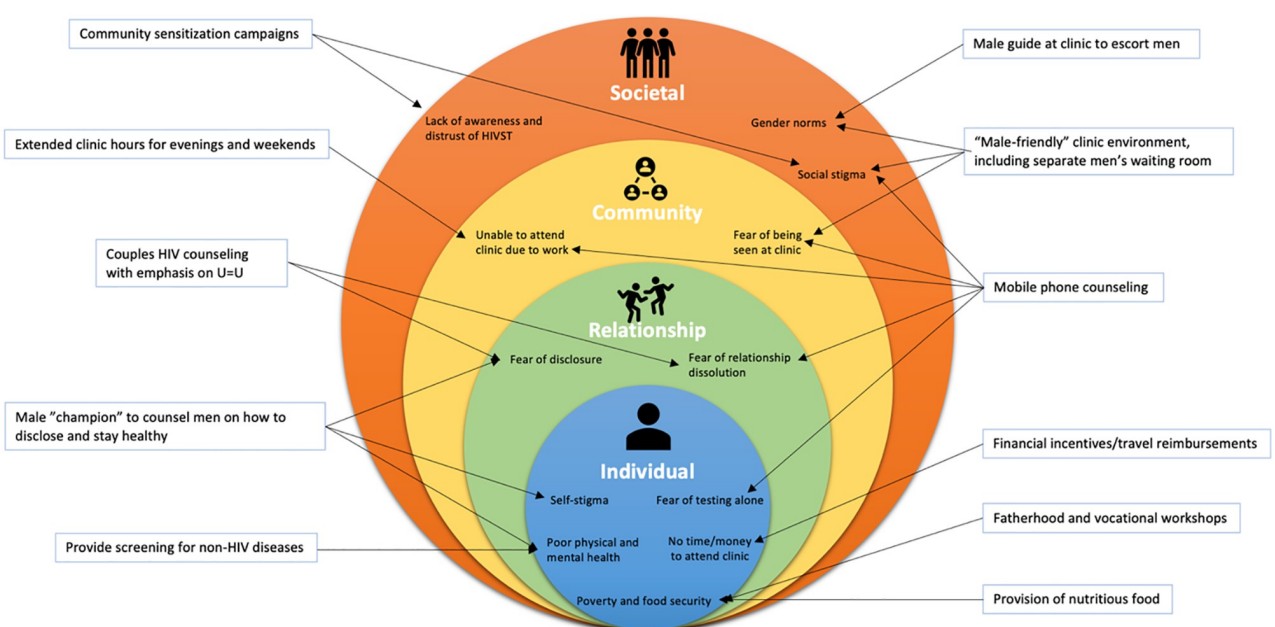

**Fig 2. Socio-ecological model with barriers and facilitators.** Fig 2 summarizes the key themes discussed by participants, based on a socio-ecological model. Within each ring of the framework, common barriers for uptake of secondary distribution of HIVST are presented. Outside of the rings, common facilitators that could support or motivate uptake are presented; arrows point to barriers that the facilitators might be relevant for addressing.

25]. FGDs and IDIs lasted approximately 90 minutes and were facilitated by a trained Ugandan qualitative researcher (JM), in discreet and convenient locations in the community, and audio-recorded. Participants were first asked about their awareness of HIVST and then provided a verbal explanation of HIVST before proceeding with remaining topics. Participants were reimbursed for time and transport (8 USD).

Recordings were transcribed verbatim and translated from Luganda to English by JM, and all personally identifiable information was removed. Data were then imported into NVivo 12 Software (QSR International, Burlington, MA) for data management and analysis. Transcripts were coded using NVivo and the coded data were analyzed by MB, BN, and MS; 20% of all FGD and IDIs were double-coded to resolve inter-coder discrepancies through team discussion and refinement of the final codebook. The coding tree intended to categorize barriers and facilitators of HIVST, and suggested strategies to encourage male uptake. Since we had a pre-defined research question, we elected to use a deductive approach based on codes and categories specified *a priori*. We used the socio-ecological model to organize key findings derived from the analysis [26] (Fig 2). This model comprises of interconnected "rings" that fit within each other: Individual, Relationship, Community, and Societal. We conducted deductive content analysis, creating a codebook guided by this framework and organizing transcripts into categories defined during coding [27]. Data from FGDs and IDIs were coded separately and combined during analysis for interpretation. Additional information about data collection and analysis, as well as a checklist of ethical, cultural, and scientific considerations regarding inclusivity in global research, can be found in the S1 File.

## Ethical approvals

This study was approved by the University of Washington (STUDY00006094), the Mildmay Uganda Research Ethics Committee (#REC REF 0112–2018) and the Uganda National

Council for Science and Technology (HS391ES). Participants provided written informed consent.

## Results

We conducted 14 FGDs (9 with men, 3 with PWHIV, one with HIV-negative pregnant women, and one with pregnant women of mixed HIV status) and 10 IDIs (5 with men and 5 with women), with a total of 122 participants (Table 1). The median age among men (N = 64) was 30 years (interquartile range [IQR] 22–54) and 27 years (IQR 19–41) among women (N = 58). Approximately 53% of women and 6.3% men self-reported having HIV. Most participants were cohabitating (87%) and another 11.5% were married. Women had higher median education than men (12th vs. 6th grade, respectively) and participants had a median of two children.

### Awareness and acceptability of HIVST and secondary distribution

**Low awareness of HIVST.**   Most participants generally reported low awareness of HIVST; often stating that they "*did not know HIV could be detected using saliva*". Some participants expressed reluctance to use a test they were not familiar with.

> "*I wouldn't want to use it [HIVST kit] because I don't understand it very well, I only know the option of removing blood from the finger.*"
>
> —*Man, age 36 years, FGD*

**Perceived benefits of HIVST.**   Once participants received an explanation about HIVST, they reported high interest in using them, mentioning HIVST's potential to overcome barriers of facility-based testing, including long travel distances, wait times, lost wages, and fear of needle pricks. Participants reported that using HIVST "*can benefit the health of the whole family*" including the unborn baby (**Table I in** S1 Appendix).

Both men and women mentioned HIVST could reach more men by overcoming concerns about confidentiality and stigma of couples- or clinic-based testing.

**Table 1.  Participant characteristics.**

|  | Overall (N = 122) | Men (N = 64) | Women (N = 58) |
|---|---|---|---|
|  | *n (%) or median (IQR)* | | |
| **Self-reported HIV status** |  |  |  |
| HIV-positive | 35 (28.7) | 4 (6.3) | 31 (53.5) |
| HIV-negative | 83 (68.0) | 57 (89.0) | 26 (44.8) |
| Don't know | 4 (3.3) | 3 (4.7) | 1 (1.7) |
| **Marital status** |  |  |  |
| Married | 14 (11.5) | 12 (18.7) | 2 (3.45) |
| Cohabitating | 106 (86.9) | 52 (81.3) | 54 (93.1) |
| Not living together | 2 (1.6) | 0 (0.00) | 2 (3.45) |
| **Has paid occupation** | 94 (77.0) | 64 (100.0) | 30 (51.7) |
| **Median age (years)** | 28 (25–32) | 30 (27–35) | 27 (24–30) |
| **Median number of children** | 2 (1–3) | 2 (1–4) | 1 (0–2) |
| **Median level of education** | 10th grade (7–12) | 6th grade (6–10) | 12th grade (9–12) |

*"HIV self-test is like a fishing net to trap men to test for HIV because men don't want to come out openly and test . . . [HIVST] is used secretly and gives safety to men who are not brave enough to go with their partners to the hospital to test."*

*HIV-negative man, age 32, FGD*

**Acceptability of HIVST secondary distribution.** Some men were supportive of secondary distribution stating that it would strengthen their relationship, since testing together "*binds the two of you*" (**Table I in** S1 Appendix). Women expressed interest in delivering HIVST kits to their partners, but only if they were in a strong relationship, characterized by "*mutual understanding.*" Women stressed the importance of receiving training from healthcare workers (HCWs) regarding HIVST use and interpretation, strategies to approach their partner so they can "*handle him properly*", including how to answer questions about why they brought HIVST to their partner.

*"You start to feel how to approach him, he may ask you, 'Why did you bring me this kit? Don't you trust me'? It needs bravery to take him the kit. . .the woman also needs to be counseled on how to handle the man so that he is likely to accept it."*

*HIV-negative woman, age 27, IDI*

## Barriers to HIVST and secondary distribution

Most HIV-negative men reported fearing HIV testing in general because an HIV-positive result "*affects you worse*" than continuing to live unaware of HIV infection. These men expressed a fatalism that HIV would mean impending death. Both men and women discussed that although men may cite lack of time or money to attend the clinic as barriers to testing, the main reason is "*the fear inside them.*" Many men reported reluctance to HIV self-test alone without counseling, particularly in the case of a positive result, stating that their "*heart can burst*" or they may harm themselves instead of going to the clinic for support. When asked about mobile phone counseling from HCWs, men stated that this would help them "*stay strong*" and "*give hope*" about linking to treatment. Meeting other men successfully living with HIV in the clinic was also described as a benefit of clinic-based testing.

A salient fear of secondary distribution was self-testing HIV-positive and disclosing to one's partner (**Table II in** S1 Appendix). Both men and women expressed fears that testing HIV-positive would lead to blame for "*bringing the virus into the relationship*". Therefore, participants hesitated to test in their partners' presence. PWHIV in particular highlighted concerns about abandonment and loss of financial support for themselves, their baby and other children due to disclosure, stating that they are "*not in a position to deliver the kit.*"

*"Remember [I] am pregnant and HIV positive. . .after self-testing he abandons you in the house and remember it is a house for renting, you have to pay landlord, look for what to eat, there are items needed in the hospital but he left you in the house and you have nowhere to look for him."*

*HIV-positive woman, age 30, FGD*

Participants commonly reported lacking confidence that adherence to ART and viral suppression can prevent HIV transmission between partners (i.e.,

"undetectable = untransmittable", "U = U"), contributing to common beliefs that serodifferent partnerships could not stay together without transmitting the virus.

> *"Am not aware about [U = U]. . .and you cannot convince me that the woman who is taking her HIV medication regularly cannot transmit HIV to me and I should have unprotected sex with her or stay with her, I don't believe it and I cannot tolerate it."*
>
> *HIV-negative man, age 48, FGD*

One man expressed that if he were diagnosed with HIV, his love for his partner would compel him to leave her to prevent spreading the virus to her and the unborn baby. Other participants reported that being in serodifferent relationships could mean staying together, but platonically, "*living as brother and sister.*" Some men shared experiences of staying in a serodifferent relationship, particularly with the use of HIV pre-exposure prophylaxis (PrEP).

> *"When I was tested positive, I realized that my wife is innocent. . . I disclosed to her and we went together [to test]. . . God answered my prayer for her to be negative. . . this helped me remain strong because I had suggested to separate with her but she insisted we stay together and the HCWs gave her drugs [PrEP] which prevent her from getting infected."*
>
> *—HIV-positive man, age 27, FGD*

Many participants expressed concerns about IPV, suggesting that secondary distribution could lead to "*quarrels,*" "*slaps*", and accusations of distrust. Several men and women stated that in relationships without "*mutual respect and understanding,*" HCWs should contact men directly and ask them to come to the facility to pick up HIVST kits instead of placing the responsibility on women to deliver kits.

In discussing gender norms, some men reported perceptions about lower intellectual capacity of women and their inability to explain HIVST use and results interpretation. Some men stated that they prefer a male HCW to call and counsel them about HIVST. Ultimately, participants agreed that a woman's ability to deliver HIVST to her partner depends on their relationship dynamics.

> *"God created a man and woman where a man's IQ is high compared to women and a man gives a listening ear to a fellow man . . .these women of ours, don't know how to explain [HIVST] to us. . .so let it be your [HCW's] responsibility."*
>
> *HIV-positive man, age 27, FGD*

Furthermore, some men expressed concerns about relying on their partner to interpret their results, fearing that she would learn his status before he does or may not tell him. They expressed reservations about sharing their HIV status with their female partners without knowing her status in return. Some participants suggested bringing home two HIVST kits would be beneficial so they can "*self-test together and each come to know one's status.*"

> *"It is upon you [the HCW] to persuade her to bring two HIVST kits and we test together, because if she refuses to use it, I too won't self-test. Instead I will give her excuses."*
>
> *HIV-negative man, age 50, FGD*

### Suggested interventions for increasing men's HIVST uptake and clinic linkage

**Community sensitization.** Most participants cited a lack of familiarity with HIVST as a barrier to their uptake. Many stressed the importance to increase awareness and confidence in one's ability to use HIVST and interpret the results. Many participants suggested it is the responsibility of HCWs to provide community sensitization "*deep in the villages*" so that everyone can "*gain confidence in its accuracy*," which would increase the likelihood of men accepting HIVST regardless of how they received it (e.g., partner, clinic, pharmacy, etc.).

**Interventions to increase confirmatory testing and clinic linkage.** Participants were asked to provide feedback about specific interventions based on the Socio-Ecological Model framework [28] (Fig 2). Participants agreed that having a male-only waiting room at the HIV clinic may increase clinic attendance, as "*men feel small in a pool of women*" (**Table III in** S1 Appendix). Some men also suggested having a male HCW call them beforehand and discreetly guide them through the clinic testing process, so they do not appear "*lost*". Participants stated that frequent follow up calls from "*vigilant HCWs*" and meeting male peer guides living healthy lives with HIV would "*give strength*" and facilitate linkage to ART.

Most men cited barriers associated with poverty, including lack of funds to attend the clinic, inability to miss work ("*a day without work is a day without food for my family*"), and concerns about starting ART if found HIV-positive because they lack adequate nutrition ("*no food in the belly to take ART with*"). Furthermore, almost all men agreed that clinics should offer extended clinic hours during evenings and weekends so they could access services during less crowded times and without missing work. Men were also asked about financial incentives, fatherhood workshops, and employment training, and most agreed such offerings would motivate them to attend the clinic.

**Perceptions of alternative HIVST distribution strategies.** Almost all participants pointed to the need for multiple options to be available (**Table IV in** S1 Appendix). Some men preferred to receive kits directly from clinics to "*ensure they are genuine*". Men's opinions regarding pharmacy-based HIVST distribution were mixed, with some participants reporting that pharmacy pick-up was discreet and convenient, similar to obtaining pregnancy tests, while others expressed mistrust in pharmacy workers to maintain confidentiality or over-charge ("*[pharmacists] are after your money*"). Views on HIVST distribution from community health workers (CHWs) were largely negative, including concerns about confidentiality (i.e. "*telling other community members that so and so was here and he tested HIV-positive*"), and fear that CHWs will try to sell them HIVST kits at elevated prices. Some men preferred phone calls from male HCWs encouraging clinic HIV testing while others expressed interests in home visits from HCWs to provide counseling and testing.

*"It is the responsibility of HCWs to get our numbers from our wives when they go for ANC services and persuade him to test, because if it was her who told me I am needed at the hospital, you would not have seen me here."*

*HIV-positive man, age 27, FGD*

**Re-envisioned cascade.** Fig 1b summarizes participants' suggested alternate strategies in a newly envisioned cascade model for secondary distribution of HIVST to improve male partner uptake.

## Discussion

Overall, participants expressed high interest in using HIVST kits themselves, but secondary distribution was not universally preferred. While participants felt secondary distribution was acceptable for couples in relationships characterized by trust and mutual understanding, PWLHIV expressed concerns regarding discovering HIV serodifference, IPV, blame, and abandonment. Additionally, both men and women cited gender dynamics as a barrier of secondary distribution. Since Ugandan men are traditionally the breadwinners and heads of households, placing women in a position of educating men about a technology could result in relationship tension or refusal of HIVST because of lack of confidence in their partners' ability to explain and interpret the test. Participants also expressed fear of testing HIV-positive without a counselor available to provide reassurance and encourage linkage to ART. Some men also expressed fear that testing HIV-positive in the presence of their partner would result in accusations of infidelity. These findings are consistent with previous studies assessing HIVST acceptability which highlight the importance of counseling availability to facilitate clinic linkage and disclosure [12, 13]. Our findings align with a recently completed randomized clinical trial in Uganda which evaluated secondary distribution of HIVST from PWHIV to their male partners and did not find an increase in men's HIV testing in the intervention arm compared to standard of care arm of clinic testing [29].

Participants' low awareness of HIVST resulted in lack of confidence in its accuracy and one's self-ability to use and interpret the test. Community sensitization of both HIVST and secondary distribution are needed to increase familiarity and demand for HIVST, reduce the need for women to educate their partners, and alleviate tensions of challenging gender norms [14]. Education is also needed to instill confidence in the ability of oral HIVST to detect HIV antibodies in the mucosal fluid despite the virus itself not being present in saliva. At the time of data collection, secondary distribution of HIVST kits to pregnant women was just starting to be scaled up in Uganda, resulting in many participants learning about HIVST for the first time through this study. Participants reported low confidence in ART adherence to prevent transmission ("Undetectable = Untransmittable"), contributing to commonly expressed beliefs that serodifferent couples could not stay together without transmitting HIV. However, a few participants shared experiences of staying together despite serodifference with the help of counseling regarding U = U and/or PrEP. Community sensitization about the potential of ART to enable individuals to live full and healthy lives without transmitting HIV to their partners or unborn children could alleviate couples' fears about HIVST while also addressing men's fear of HIV testing in general. When offering secondary distribution of HIVST to pregnant women, midwives and counselors may need to provide additional counseling time for PWLHIV to successfully deliver HIVST to their partners, compared to pregnant women without HIV who do not face the same fears around disclosure and abandonment.

Our findings highlight opportunities to improve HIVST delivery. Providing men alternative avenues for obtaining HIVST can avoid the burden placed on pregnant women of being solely responsible for HIVST distribution, particularly for PWHIV, women at risk of IPV, or those in unstable partnerships. Pregnancy is a high-risk time for women in SSA since relationship dissolution could lead to economic vulnerability for themselves and their children. Additionally, pregnant women are at high risk of IPV and HIV acquisition, and PWHIV experiencing IPV are at higher risk of treatment non-adherence [30, 31]. According to the Ugandan Demographic and Health Survey, 39.6% of women ≥15 years had experienced physical, sexual, or psychological IPV within the last 12 months [32]. HIVST secondary distribution has not been evaluated in the context of high IPV risk and therefore may not currently be

appropriate for women with IPV risk. We suggest a hybrid model (Fig 1b) which involves providing females the option of having male HCWs call their partners directly to offer the option of coming to the facility to obtain HIVST. Regardless of distribution strategy, participants expressed desire for phone-based or in person counseling from a male HCW after HIVST. Linkage strategies to promote men's clinic attendance after HIVST are not well evaluated. When asked to provide feedback on specific interventions, men stated that separate "male friendly" waiting rooms, male guides at the clinic, and male "champions" (men living with HIV who counsel newly diagnosed men) may encourage clinic linkage. Lastly, extended clinic hours, financial incentives, and workshops could address the lack of time and money to miss work. Our findings align with previous studies which find that providing men with financial incentives can increase engagement with HIV services [33, 34]. Further research is needed on the potential of other suggested interventions to increase men's linkage.

Our study has several limitations. Participants were purposefully sampled and may not represent the views of all pregnant women and male partners. We enrolled men whose female partners consented to provide their contact information, which may bias our sample towards those in stable partnerships. Few participants reported experience with HIVST secondary distribution, but most discussed their preferences hypothetically, which may not correlate with health behavior. The FGDs may be subject to social desirability bias. However, we stratified FGDs by gender and HIV status and conducted IDIs to further elicit participant views. Further, our qualitative interviewer was male; while this was likely a strength for obtaining male participant perspectives, it could have impacted women's comfort in discussing sensitive topics such as HIV, sexual behavior, and gender roles.

Strengths of our study include recruiting men not accompanying their partners to ANC and oversampling PWHIV. Our findings differ from previous qualitative studies on HIVST secondary distribution which found high acceptability among participants [24, 35, 36]. One study recruited men who were attending ANC, who may be more likely to be in stable partnerships conducive to HIVST delivery [24]. Another study recruited pregnant women and male partners who successfully used HIVST, thus not capturing the perspectives of women who did not deliver HIVST and men who refused to self-test [35]. These studies do not capture barriers that prevented successful HIVST delivery or testing uptake. Additionally, men attending ANC may be more likely to undergo clinic HIV testing. Assessing perspectives of men not attending ANC with their partners provides important information on the potential of HIVST for men who do not test at the clinic. Finally, the third study did not purposively sample PWHIV [36], who experience distinct barriers and will require targeted counseling to enable successful delivery of HIVST and reduce risks of adverse events. Although we found high acceptability for collecting HIVST from clinics, perspectives on pharmacy and CHW-distribution were largely negative; participants described lacking trust in providers, concerns about confidentiality and counterfeit or over-priced tests. Our finding differs from previous studies showing high acceptability of pharmacy-based HIVST distribution in Kenya [37]. However, that study sampled participants who expressed interest in HIVST so may not be representative of population acceptability.

## Conclusion

Overall, we find providing participants alternative avenues for HIVST distribution and options for counseling support may increase uptake among male partners of pregnant women in Uganda and similar settings. Future studies are needed to assess the potential of integrating tailored strategies into existing programs to safely expand HIV testing and clinic linkage among men.

## Supporting information

**S1 File.**
(PDF)

**S1 Appendix. Additional quotes.**
(DOCX)

## Acknowledgments

We would like to thank the study team and the study participants. We gratefully acknowledge Dr. Karusa Kiragu, the Country Director of The Joint United Nations AIDS Program (UNAIDS) in Uganda, for providing her insight and expertise.

## Author Contributions

**Conceptualization:** Michelle A. Bulterys, Andrew Mujugira, Jackson Mugisha, Agnes Nakyanzi, Faith Naddunga, Jade Boyer, Norma Ware, Connie Celum, Monisha Sharma.

**Data curation:** Michelle A. Bulterys, Andrew Mujugira, Jackson Mugisha, Agnes Nakyanzi.

**Formal analysis:** Michelle A. Bulterys, Brienna Naughton, Andrew Mujugira, Jackson Mugisha, Norma Ware, Monisha Sharma.

**Funding acquisition:** Michelle A. Bulterys, Norma Ware, Connie Celum, Monisha Sharma.

**Investigation:** Michelle A. Bulterys, Andrew Mujugira, Jackson Mugisha, Agnes Nakyanzi, Norma Ware, Connie Celum, Monisha Sharma.

**Methodology:** Michelle A. Bulterys, Andrew Mujugira, Norma Ware, Connie Celum, Monisha Sharma.

**Project administration:** Michelle A. Bulterys, Jackson Mugisha, Agnes Nakyanzi, Faith Naddunga, Jade Boyer, Connie Celum, Monisha Sharma.

**Resources:** Michelle A. Bulterys, Andrew Mujugira, Norma Ware, Connie Celum, Monisha Sharma.

**Software:** Michelle A. Bulterys, Monisha Sharma.

**Supervision:** Michelle A. Bulterys, Andrew Mujugira, Agnes Nakyanzi, Faith Naddunga, Jade Boyer, Norma Ware, Connie Celum, Monisha Sharma.

**Validation:** Michelle A. Bulterys, Andrew Mujugira, Norma Ware, Connie Celum, Monisha Sharma.

**Visualization:** Michelle A. Bulterys, Jade Boyer, Connie Celum, Monisha Sharma.

**Writing – original draft:** Michelle A. Bulterys, Monisha Sharma.

**Writing – review & editing:** Michelle A. Bulterys, Brienna Naughton, Andrew Mujugira, Jackson Mugisha, Agnes Nakyanzi, Faith Naddunga, Jade Boyer, Norma Ware, Connie Celum, Monisha Sharma.

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
