## [Decision Letter · Decision Letter 0]

10 Aug 2022

PONE-D-22-17033Pregnant women and male partner perspectives of secondary distribution of HIV self-testing kits in Uganda: A qualitative studyPLOS ONE

Dear Dr. Bulterys,

Thank you for submitting your manuscript to PLOS ONE. After careful consideration, we feel that it has merit but does not fully meet PLOS ONE’s publication criteria as it currently stands. Therefore, we invite you to submit a revised version of the manuscript that addresses the points raised during the review process.

We look forward to receiving your revised manuscript.

Kind regards,

Joel Msafiri Francis, MD, MS, PhD

Academic Editor

PLOS ONE

Journal Requirements:

"NO authors have competing interests"

Reviewers' comments:

Reviewer's Responses to Questions

**Comments to the Author**

1. Is the manuscript technically sound, and do the data support the conclusions?

Reviewer #1: Yes

Reviewer #2: Partly

2. Has the statistical analysis been performed appropriately and rigorously? 

Reviewer #1: N/A

Reviewer #2: N/A

3. Have the authors made all data underlying the findings in their manuscript fully available?

Reviewer #1: Yes

Reviewer #2: Yes

4. Is the manuscript presented in an intelligible fashion and written in standard English?

Reviewer #1: Yes

Reviewer #2: Yes

5. Review Comments to the Author

Reviewer #1: Pregnant women and male partner perspectives of secondary distribution of HIV self-testing kits in Uganda: A qualitative study.

Congratulations on your study and development of this manuscript for publication. A study of this nature potentially can add to the body of knowledge on strategies to increase uptake of HIV testing among men in SSA. Below I offer comments that could make the manuscript more improved and valuable.

GENERAL COMMENT:

Overall the study was well designed and implemented and the manuscript is well written.

However I have few comments below to improve the manuscript.

1. Abstract: I would like to suggest that the authors should be consistent in using the words:“ Facilitators vs. motivators”. These words are used interchangeable through out the manuscript. See line 62 on page 3, line 112 on pages 5, and line 122 on pages 6, etc.

2. Introduction: The following sentence mentions “…regarding HIVST acceptability, barriers….”. Did the authors also assessed ”Motivators/facilitators?. If yes, it is not mentioned. See line 96 page 5.

3. Data collection and analysis: Did JM conduct all the interviews alone? What about a note taker? See line 125 on page 6.

4. It seems JM (male interviewer) conducted interviews for women. Could this be a limitation for women to express themselves freely on such sensitive issues to a male interviewer compared to a female interviewer? Could authors clarify on this observation? See line 125 on page 6.

5. In principal translations should be done by a qualified translated. Does this imply that JM is a qualified Translator? Could authors clarify on this observation? See line 129 on page 6.

6. The authors should mention what was the role of using the NVivo software in this study. See line 130 on page 6.

7. The authors mentioned that all FGDs and IDIs were double-coded to assess inter-coder reliability. Could authors provide the results of the inter-coder reliability? See 131 on page 6.

8. The authors also mentioned that they used the Social ecological model to organize the research questions and findings. However, It was very difficult to follow how the authors operationalize their findings and what was presented in Figure II. I suggest that the authors should align their findings and what is presented in the current Figure II. An additional description accompanying the Figure II will be very beneficial to readers. See line 132 on page 6 and line 170 on page 9.

9. The authors reported that they used deductive thematic analysis. The authors should provide justification of using the deductive thematic analysis compared to inductive thematic analysis. See line 134 on page 6.

10. The authors did not mention whether FGDs and IDIs data were analyzed jointly or separately and combined later during interpretation.

11. The authors did not mention how they applied reflexivity and rigor in terms of trustworthness, dependability, credibility and transferability throughout the data analysis from data collection to interpretation of the findings.

12. Ethical approval: The authors should provide ethics clearance certificate ID for both Ethics committees. See line 137 on page 6.

13. Authors did not mention measures taken to protect confidentiality and anonymity in this study.

14. Results: It was unnecessary to present median & IQR in a qualitative study. Could authors clarify on this observation? See line 144 and 148 on page 7.

15. The following sentence….”did not know HIV could be detected using saliva”. I think using the word ‘saliva’ is not correct in the HIVST perspective. What is used is oral fluid from the mucosa of the mouth. I suggest this should be corrected. See line 176 on page 9.

16. The following quote: “ HIV self-test is like a fishing net…their partners to the hospital to test”. This quote does not support what is described above. Authors should justify the use of the quote or present a relevant quote. See lines 187-190 on page 10.

17. Conclusion: Is missing in the main manuscript but presented in the abstract. Authors should add their conclusion in the main manuscript.

18. Authors have mentioned among other interventions “ financial incentives”. Despite existing literature supporting this intervention, I was wondering if this would be sustainable in case of scaling-up to the general population.

19. One of the key barriers for uptake of HIVST is the high cost of buying the HIV self-test kits. I was wondering if the authors assessed this common barrier.

Reviewer #2: Comments to the Author

Summary: Overall these study findings add very little evidence on secondary distribution of HIVST kits in an ante natal setting. I would recommend that the authors made major revisions. These include contexualising how their findings are different from other studies conducted in other countries in Africa.

Detailed comments are below.

Abstract:

1. Specify with whom FGDs and IDI’s were conducted (men, women etc)

Introduction:

1. I recommend providing stats on HIV testing in Uganda.

2. Briefly describe what testing is like among men and women.

3. Need to specify what the current status of HIVST in Uganda from a policy perspective.

4. The last paragraph is not clear what you are trying to establish. Many studies have already been done on acceptability and barriers. Many conducted on secondary distribution within ANC care. How is yours different?

Methods:

1. How was low risk for IPV established?

2. How many FGDs, how many IDI’s.

3. There are a lot of elements from the Consolidated Criteria for Reporting Qualitative Studies (COREQ) missing from the methods (https://academic.oup.com/intqhc/article/19/6/349/1791966). I strongly encourage the authors to consult the COREQ criteria and ensure that elements are covered

4. For each ‘finding’ (new paragraph), it would be preferable to provide some sense of how common it was, and among whom (women, men,). Doesn’t have to be as specific as the “Most (85%)…” under the first finding (although that level of specificity is helpful there). This will give a better sense, at the end, for a sort of pros/cons assessment which is currently missing – high acceptability and perceived effectiveness, and low concerns? Or more equal levels of both?

Results:

1. Information in first paragraph should be in the methods section

2. Where the men and women interviews also held by status? Refer to 2-3 line of first paragraph under methods.

3. Under “low awareness of HIVST”, there is mention of saliva. Context needs to be provided in the introduction and discussion of HIVST in Uganda, including tests used.

Discussion:

1. Results are comparative of other studies. How is this study different?

2. Most of the discussion summarises the results. Doesn’t offer much in terms of the strategies identified could be integrated.

3. To posit this work as a formative qualitative study, more information is needed to contextualize why this study was specifically needed, how it informed intervention development, and how the newly formed intervention would be subsequently used. qualitative study.

6. PLOS authors have the option to publish the peer review history of their article (what does this mean?). If published, this will include your full peer review and any attached files.

Reviewer #1: No

Reviewer #2: **Yes: **Angela Tembo

---

## [Author Response · Author response to Decision Letter 0]

17 Oct 2022

Thank you for the opportunity to revise and resubmit our manuscript. Our responses to reviewer comments are attached.

---

## [Decision Letter · Decision Letter 1]

14 Dec 2022

Pregnant women and male partner perspectives of secondary distribution of HIV self-testing kits in Uganda: A qualitative study

PONE-D-22-17033R1

Dear Dr. Bulterys,

We’re pleased to inform you that your manuscript has been judged scientifically suitable for publication and will be formally accepted for publication once it meets all outstanding technical requirements.

Kind regards,

Joel Msafiri Francis, MD, MS, PhD

Academic Editor

PLOS ONE

Additional Editor Comments (optional):

Reviewers' comments:

Reviewer's Responses to Questions

**Comments to the Author**

1. If the authors have adequately addressed your comments raised in a previous round of review and you feel that this manuscript is now acceptable for publication, you may indicate that here to bypass the “Comments to the Author” section, enter your conflict of interest statement in the “Confidential to Editor” section, and submit your "Accept" recommendation.

Reviewer #1: All comments have been addressed

Reviewer #2: All comments have been addressed

2. Is the manuscript technically sound, and do the data support the conclusions?

Reviewer #1: Yes

Reviewer #2: Yes

3. Has the statistical analysis been performed appropriately and rigorously? 

Reviewer #1: Yes

Reviewer #2: N/A

4. Have the authors made all data underlying the findings in their manuscript fully available?

Reviewer #1: Yes

Reviewer #2: Yes

5. Is the manuscript presented in an intelligible fashion and written in standard English?

Reviewer #1: Yes

Reviewer #2: Yes

6. Review Comments to the Author

Reviewer #1: I am pleased that the authors have addressed all comments that I suggested and the manuscript can be accepted for publication.

Reviewer #2: Dear Editor,

The authors have adequately addressed the comments. The paper is now of publishable standard. I would however recommend that the authors consider repositioning the “re-envisioned cascade” to recommendations section.

Thank you

7. PLOS authors have the option to publish the peer review history of their article (what does this mean?). If published, this will include your full peer review and any attached files.

Reviewer #1: No

Reviewer #2: **Yes: **Angela Tembo

---

## [Editor Report · Acceptance letter]

2 Feb 2023

PONE-D-22-17033R1 

Pregnant women and male partner perspectives of secondary distribution of HIV self-testing kits in Uganda: A qualitative study 

Dear Dr. Bulterys:

I'm pleased to inform you that your manuscript has been deemed suitable for publication in PLOS ONE. Congratulations! Your manuscript is now with our production department. 

Kind regards, 

on behalf of

Dr. Joel Msafiri Francis 

Academic Editor

PLOS ONE